# Associations between Lifestyle Factors and Vitamin E Metabolites in the General Population

**DOI:** 10.3390/antiox9121280

**Published:** 2020-12-15

**Authors:** Leon G. Martens, Jiao Luo, Fleur L. Meulmeester, Nadia Ashrafi, Esther Winters van Eekelen, Renée de Mutsert, Dennis O. Mook-Kanamori, Frits R. Rosendaal, Ko Willems van Dijk, Kevin Mills, Raymond Noordam, Diana van Heemst

**Affiliations:** 1Department of Internal Medicine, Section of Gerontology and Geriatrics, Leiden University Medical Center, Albinusdreef 2, 2333 ZA Leiden, The Netherlands; l.g.martens@lumc.nl (L.G.M.); j.luo@lumc.nl (J.L.); f.l.meulmeester@lumc.nl (F.L.M.); d.van_heemst@lumc.nl (D.v.H.); 2Department of Clinical Epidemiology, Leiden University Medical Center, Albinusdreef 2, 2333 ZA Leiden, The Netherlands; e.van_eekelen@lumc.nl (E.W.v.E.); r.de_mutsert@lumc.nl (R.d.M.); d.o.mook@lumc.nl (D.O.M.-K.); f.r.rosendaal@lumc.nl (F.R.R.); 3NIHR Great Ormond Street Biomedical Research Centre, Great Ormond Street Hospital and UCL Great Ormond Street Institute of Child Health, 30 Guilford St, Holborn, London WC1N 1EH, UK; nadia.ashrafi@ucl.ac.uk (N.A.); kevin.mills@ucl.ac.uk (K.M.); 4Department of Public Health and Primary Care, Leiden University Medical Center, Albinusdreef 2, 2333 ZA Leiden, The Netherlands; 5Department of Human Genetics, Leiden University Medical Center, Albinusdreef 2, 2333 ZA Leiden, The Netherlands; k.willems_van_dijk@lumc.nl; 6Department of Internal Medicine, Division of Endocrinology, Leiden University Medical Center, Albinusdreef 2, 2333 ZA Leiden, The Netherlands

**Keywords:** vitamin E, antioxidants, lifestyle, smoking, sleep, physical activity, alcohol, diet

## Abstract

The antioxidant vitamin E (α-tocopherol, α-TOH) protects lipids from oxidation by reactive oxygen species. We hypothesized that lifestyle factors associate with vitamin E metabolism marked by urinary α-tocopheronolactone hydroquinone (α-TLHQ) and α-carboxymethyl-hydroxychroman (α-CEHC levels), as potential reflection of lipid oxidation. We conducted a cross-sectional study in the Netherlands Epidemiology of Obesity Study. Serum α-TOH, and urinary α-TLHQ and α-CEHC were quantified by liquid chromatography coupled with tandem mass spectrometry. Information on the lifestyle factors (sleep, physical activity (PA), smoking and alcohol) were collected through questionnaires. Multivariable linear regression analyses were performed to assess the associations between the lifestyle factors and α-TOH measures. A total of 530 participants (46% men) were included with mean (SD) age of 56 (6) years. Of the examined lifestyle factors, only poor sleep was associated with a higher serum α-TOH (mean difference: 4% (95% CI: 1, 7%)). Current smoking was associated with higher urinary α-CEHC (32%: (14%, 53%)), with evidence of a dose–response relationship with smoking intensity (low pack years, 24% (2, 52%); high pack years, 55% (25, 93%)). Moderate physical activity was associated with a lower α-TLHQ relative to α-CEHC (−17%: (−26, −6%), compared with low PA). Only specific lifestyle factors associate with vitamin E metabolism. Examining serum α-TOH does not provide complete insight in vitamin E antioxidant capacity.

## 1. Introduction

The lipid-soluble antioxidant vitamin E is a defensive compound neutralizing reactive oxygen species (ROS), specifically when formed during lipid peroxidation [1]. The most common form of vitamin E in the blood is α-tocopherol (α-TOH) [2]. During the process of neutralizing lipid peroxidation reactions, α-TOH can be metabolized through two different mechanisms, notably enzymatic degradation and oxidation. When α-TOH is engaged in an oxidative reaction, the chromanol ring is opened which results in the formation of the oxidation product α-tocopheronolactone hydroquinone (α-TLHQ) [1,3,4]. The remaining is degraded through enzymatic conversion into α-carboxymethyl-hydroxychroman (α-CEHC), and is a measure of α-TOH status. Once formed, both metabolites are processed in the liver and excreted via the urine [4].

In most published studies, an individual’s vitamin E status is reflected by the measured α-TOH level in the blood serum. In these studies, increased levels of α-TOH have been associated with reduced risks of age-associated diseases [5]. A meta-analysis showed that dietary intake of the antioxidants vitamin E, vitamin C and β-carotene were associated with a lower risk on Alzheimer’s disease [6]. Furthermore, a study done on 39,910 middle-aged men observed a lower risk of coronary heart disease among men with a higher vitamin E intake [7]. However, recent studies have suggested that serum α-TOH does not correlate with vitamin E antioxidant activity nor with vitamin E metabolism [8]. Although based on observational data there seems to an association between high vitamin E intake and a decreased risk of cardiovascular disease CVD, clinical trials involving vitamin E supplementation have failed to show an effect on the prevention of CVD [9] and other age-related diseases [10,11,12,13,14,15]. Instead, there is increasing evidence that measuring vitamin E metabolites (notably CEHC and TLHQ) in urine provides a more accurate and reliable estimate of the vitamin E antioxidant status in the body [16]. However, it remains to be elucidated what factors influence vitamin E metabolite concentration.

A particular candidate for factors that affect vitamin E metabolism is lifestyle, which is generally accepted to have a serious impact on health [17,18,19,20,21]. For example, large studies have shown that smoking is associated with an increased the risk of diabetes mellitus and cardiovascular events [22,23]. Furthermore, it is well-established that an increase in physical activity results in clinically relevant health benefits [24]. One potential mechanism linking lifestyle to health may be reactive oxygen species (ROS) production or antioxidant activity [1]. As lifestyle factors such as smoking and physical activity are strongly linked to lipid peroxidation, and vitamin E is the main antioxidant that confers protection against lipid peroxidation-induced damage [25,26], we hypothesized that lifestyle factors are associated with vitamin E metabolite levels. Indeed, multiple of the main lifestyle factors have been studied previously in relation to serum α-TOH, but evidence is lacking for the vitamin E metabolite levels [27,28]. Previously, such hypothesis has already been postulated for nutrition [29].

In this study, we measured serum α-TOH levels in fasting serum samples and vitamin E metabolites in 24-h urine and aimed to examine associations of the main lifestyle factors smoking, sleep, physical activity and habitual food and alcohol intake with these levels in a cross-sectional cohort of middle-aged individuals.

## 2. Materials and Methods

### 2.1. Setting and Study Design

The Netherlands Epidemiology of Obesity (NEO) study is a population-based, prospective cohort study designed to investigate pathways that lead to obesity-related diseases. The NEO study started in 2008 and includes 6671 individuals aged 45–65 years, with an oversampling of individuals being overweight or obese. The study design and population has been described in detail elsewhere [30]. The Medical Ethical Committee of the Leiden University Medical Center (LUMC) approved the design of the study. All participants gave their written informed consent.

In short, men and women living in the greater area of Leiden (in the west of the Netherlands) were invited by letters sent by general practitioners, municipalities and by local advertisements. They were invited to respond if they were aged between 45 and 65 years and had a self-reported Body Mass Index (BMI) of 27 kg/m^2^ or higher. In addition, all inhabitants aged between 45 and 65 years from one municipality (Leiderdorp) were invited to participate irrespective of their BMI. This resulted in a population of 1671 participants with a BMI distribution similar to that of the general population [30].

Participants were invited to a baseline visit at the NEO study center of the LUMC after an overnight fast. Prior to this study visit, participants collected their urine over 24 h and completed a general questionnaire at home to report demographic, lifestyle and clinical information, habitual food intake and physical activity. The participants were asked to bring all medication they were using in the month preceding the study visit and names and dosages of all medication were recorded. At the baseline visit, several measurements were performed including anthropometry and blood pressure, and blood sampling after an overnight fast.

For the present analysis, we selected participants from the Leiderdorp municipality comprising of a Caucasian Dutch population with no BMI requirements for participation. At the baseline visit, participants completed a screening form to identify contra-indications for undergoing magnetic resonance imaging (MRI) (most notably metallic devices, claustrophobia or a body circumference of more than 1.70 m). Of the participants who were eligible, approximately 40% were randomly selected to undergo MRI. All individuals with available urine collected for more than 20 h were included for vitamin E metabolites measurements (*n* = 539). Individuals that had unrealistic levels of vitamin E metabolites were excluded (*n* = 9).

### 2.2. Lifestyle Factors

All lifestyle exposures were collected through self-reported questionnaires. Smoking was defined as current, former and never smoking. Long-term tobacco exposure was expressed in pack years of smoking, calculated by multiplying the number of packs of cigarettes smoked per day by the number of years the person smoked. One pack year is defined as twenty cigarettes smoked every day for one year. Additionally, to limit potential measurement error and have an objective measure for smoking exposure, we performed analyses on the relation between the metabolite cotinine and urinary vitamin E metabolites. Cotinine, a xenobiotic metabolite for nicotine, as measured using untargeted metabolomics measurements at Metabolon Inc. (Durham, NC, USA) using their Metabolon™ Discovery HD4 platform. In brief, this process involves four independent ultra-high-performance liquid chromatography mass spectrometry (UHPLC-MS/MS) platforms [31,32]. Two platforms used positive ionization reverse phase chromatography, one used negative ionization reverse phase chromatography, and one used hydrophilic interaction liquid chromatography (HILIC) negative ionization [32]. Diet quality was measured with the Dutch Healthy Diet Index (DHD-index), which measures the adherence to the Dutch dietary guidelines using a semiquantitative self-administered 125-item Food Frequency Questionnaire (FFQ) [33]. For the present study, we used an adapted version of the DHD-index with thirteen components instead of the original fifteen because we were not able to estimate the two components consumption of unfiltered coffee, and of sodium on the basis of the FFQ used in our study. As a result, the DHD-index in our study ranges between 0 and 130, where a higher score indicates a better diet quality. Sleep quality was measured with the Pittsburgh Sleep Quality Index (PSQI) questionnaire [34]. This questionnaire consists of 19 items with a total score of 21, where a higher score indicates a worse sleep quality. Alcohol consumption was estimated using the FFQ and restructured to the unit of g per day. Participants reported the frequency and duration of their usual physical activity during leisure time in the Short Questionnaire to assess health-enhancing physical activity (SQUASH), which was expressed in h per week of metabolic equivalents [35].

### 2.3. Other Variables

Body weight was measured and percent body fat was estimated by the Tanita bio impedance balance (TBF-310, Tanita International Division, Manchester, UK) without shoes and 1 kg was subtracted to correct for weight of clothing. BMI at baseline was calculated by dividing the weight in kg by the height in meters squared. LDL-cholesterol concentration was estimated using the Friedewald formula [36].

### 2.4. Serum and Urinary Vitamin E Metabolites

Fasting blood samples were obtained from the antecubital vein. Standard clinical laboratory parameters including glucose and insulin concentrations and lipid profile were obtained in the clinical chemistry laboratory of the LUMC. The remaining blood samples were aliquoted and stored at 80 °C for future research. Both measures were performed in 2019 making the storage period similar. Serum α-tocopherol levels were obtained using untargeted LC-MS/MS (Metabolon, Inc., Durham, NC, USA), as highlighted in more detail above [37].

Prior to the study visit, participants collected urine over 24 h and recorded the time of the first and last void. In addition, participants collected the first morning spot on the day of their study visit. The 24-h Urinary oxidized α-TOH metabolites (α-TLHQ) and enzymatic metabolites (α-CEHC), presented as their sulfate or glucuronide conjugates (α-TLHQ-SO3, α-TLHQ-GLU, α-CEHC-SO3, α-CEHC-GLU), were measured by LC-MS/MS at University College London, UK.

Urine samples were thawed, and 100 μL fresh urine was then centrifuged in Eppendorf tubes for 10 min at 13,000 rpm at room temperature and spiked with 10 μL of the internal standards (100 μmol/L), lithocholic acid sulfate (LA) and androsterone D4-glucuronide (AD4). Subsequently, samples were vortexed and transferred into screw-cap glass vials. Then, 10 μL was injected into the LC-MS/MS for detection.

The metabolites were separated using a Waters ACQUITY UPLC BEH C8 column (1.7 μm particles, 50 mm x 2.1 mm; Waters Corp, Manchester, UK) plus a guard column containing an identical stationary phase. The mobile phase was a gradient elution of solvent A (99.98% water; 0.01% (*v*/*v*) formic acid) and solvent B (99.98% acetonitrile/MeCN; 0.01% (*v*/*v*) formic acid), which were LC-MS grade or equivalent (Sigma-Aldrich Co. Ltd., St. Louis, MO, USA). The flow rate was set to 0.8mL/min and the LC gradient was established by coordinating the solvents as follows: 95% solvent A plus 5% solvent B for 0 to 0.40 min; 80% solvent A plus 20% solvent B for 2 min; 0.1% solvent A plus 99.9% solvent B for 3.01 to 4 min; 95% solvent A plus 5% solvent B for 4.01 to 5 min. In order to minimize system contamination and carry over, the MS diverter valve was set up to discard the UPLC eluent before and after the sample elution, at 0 to 0.40 min and 4.01 to 5 min, respectively, as well as an additional run of blank sample (H2O: MeCN) between each run of urine samples. Two peaks were observed for α-TLHQ and α-CEHC glucuronide conjugates, corresponding to major and minor isoforms.

After separation, the metabolites were then analyzed by MS using a Waters ACQUITY UPLC (Milford, MA, USA) coupled to a triple-quadrupole Xevo TQ-S fitted with an electrospray ionization in negative ion mode. The gas temperatures persisted 600 °C for desolvation. In addition, nitrogen was used as the nebulizing gas with 7.0 Bar. The cone voltages were set at 56 V and 54 V, and the collision voltages at 28 eV and 30 eV for sulfate conjugates and glucuronide conjugates, respectively. Running time for each sample is 5 min with a 20 μL injection volume together with a partial loop with needle overfill mode. Using multiple reaction monitoring (MRM) mode, specific parent and daughter ions were determined in scan mode and following collision activated dissociation (CAD) with argon. These ions were then used to quantify each α-TOH metabolite from transitions (α-TLHQ-GLU, 453.3 > 113.0; α-TLHQ-SO3, 357.1 > 79.9; α-CHE-GLU, 453.2 > 113.0; α-CHE-SO3, 357.1 > 79.9) that corresponded to their theoretical molecular masses.

Urinary creatinine concentrations (mmol/L) were also measured to correct for dilution differences for each metabolite, by triple-quadrupole Micro Quattro mass spectrometry (MicroMass, Waters, Wilmslow, UK) using deuterated creatinine as the internal standard. Therefore, the concentrations of α-TOH metabolites are expressed as nmol per mmol of creatinine. A quality control (QC) assessment was performed throughout the quantification both in creatinine and α-TOH metabolite assays to deal with the variations in sample quality and UPLC-MS/MS performance over time. Four QC samples were systematically interleaved every 50 urine samples to limit the amount of sample loss. The whole measurement protocol was developed and further modified by the detection group in London [37,38].

The final concentration of glucuronide conjugates for α-TLHQ and α-CEHC were the sum of their corresponding major and minor isoforms. In addition to the measured single metabolite, total, glucuronide and sulfate conjugates ratios were further determined to reflect the α-TOH antioxidative capacity as well as lipid peroxidation levels taking α-TOH status into consideration, namely α-TLHQ-to-α-CEHC ratio, α-TLHQ-GLU-to-α-CEHC-GLU ratio and α-TLHQ-SO3-to-α-CEHC-SO3 ratio.

### 2.5. Statistical Analysis

We examined the characteristics of the total study population and stratified by sex. Characteristics are presented as mean (standard deviations), medians (with interquartile ranges; skewed variables only) and proportions. Serum α-TOH, and urinary α-TLHQ and α-CEHC had a skewed distribution and were natural log-transformed. To assess the correlation between α-TOH, α-TLHQ and α-CEHC, we performed a Pearson correlation.

Participants with a PSQI score of 5 or higher, were assigned to the poor sleep quality group. The group of participants with a score lower than 5 were assigned to the good sleep quality group and considered the reference. For smoking, participants were distributed into three groups: (i) non-smokers (reference), (ii) former smokers and (iii) current smokers. For both diet quality, alcohol use and physical activity, the group was divided into quartiles, with the lowest quartile as the reference.

For the analyses, we examined the associations between each individual lifestyle factor and vitamin E metabolite levels using multivariable linear regression analyses adjusted for age and sex. Additionally, we adjusted for BMI and we also performed a sensitivity analysis where we substituted BMI for total body fat. As results were not different between adjusting for BMI and total body fat, we only reported results adjusted for BMI. To study the possible association between smoking cessation on α-TOH and its metabolites among former smokers, we additionally stratified the group of former smokers based on the duration of smoking cessation (less or more than 5 years of smoking cessation). As we did not observe a difference in levels of α-TOH and its metabolites, we treated the group of former smokers as one in all analyses. As lifestyle factors are likely to be considerably different between men and women, we repeated all analyses after stratifying by sex. After the analyses, all beta estimates and 95% confidence limits were back transformed to be able to present these as a percentage difference. Consequently, all results can be interpreted as the percentage difference in outcome compared with the reference category, with the 95% confidence interval.

## 3. Results

### 3.1. Characteristics of the Study Population

After excluding the nine outliers, a total of 530 participants (46% men) were analyzed. The mean (SD) age was 55.9 (6.0), and BMI 25.9 (4.0). The baseline characteristics are presented in Table 1. Among current smokers, men had smoked more in pack years than women (median: 26.6, IQR: 14.0, 36.4 versus median: 13.3, IQR: 4.7, 22.7). Men also consumed more alcohol (g/day) (median: 16.7, IQR: 5.1, 28.4 versus median: 7.2, IQR: 1.0, 14.4). α-TLHQ and α-CEHC were correlated with a Pearson correlation of 0.70. α-TLHQ and α-TOH had a Pearson correlation of 0.24. α-CEHC and α-TOH had a Pearson correlation of 0.21.

### 3.2. Association between Smoking (Intensity) and Vitamin E (Metabolite) Levels in Serum and Urine

The association between smoking and vitamin E serum and urinary metabolites is displayed in Table 2 and Appendix A. After adjusting for age, sex and potential confounding factors such as BMI and alcohol consumption, there was no evidence for a difference in mean α-TOH (−3% Beta (95% CI: −8, 2)) level between current smokers and never-smokers. For the urinary α-TOH metabolites, current smoking was associated with a 32% lower α-CEHC (−32% Beta (95% CI: −44, −18)) and a 32% higher TLHQ relative to CEHC ratio (32% Beta (95% CI: 14, 53)), after adjustments for possible confounding. In addition, we observed no difference in mean levels of α-TOH or its metabolite levels between never smokers and past smokers. An increased smoking intensity (measured in pack years, and restricted to never and current smokers) was associated with lower α-TOH levels, with current smokers grouped in the high pack years group having a 10% (95% CI: −16, −3)) lower α-TOH level than never smokers (Figure 1D). For urinary vitamin E metabolites, both low and high pack years were associated with reduced α-CEHC (Figure 1A) and an increased TLHQ relative to CEHC ratio (Figure 1C, Appendix A), in a clear dose–response relationship. Notably, α-CEHC was 25% (95% CI: −43, −2) lower in current smokers with low pack years and 46% (95% CI: −59, −28) lower in current smokers with high pack years, than in never smokers. The TLHQ relative to CEHC ratio was 24% (95% CI: 2, 52) higher in current smokers with low pack years and 55% (95% CI: 25, 93) higher in current smokers with high pack years, than in never smokers.

As an objective measure of smoking status, the presence of cotinine in serum was associated with lower α-CEHC (difference: −20%, 95% CI: −30, −8). Furthermore, after splitting the cotinine present group in two halves, the group with the highest cotinine level had a significantly lower α-CEHC (difference: −28%, 95% CI: −40, −14) when compared with the reference group (Table 3).

### 3.3. Diet Quality and Vitamin E Serum and Urinary Levels

There was no association between the DHD-index and levels of vitamin E or its metabolites, (Table 2 and Appendix A). For the fourth quartile, the level of α-TOH was not different (2% (95% CI: −2, 7)) from what was observed in those grouped having the worst diet quality. A similar result was observed for the vitamin E metabolites α-TLHQ (12% (95% CI: −3, 29)), α-CEHC (10% (95% CI: −7, 31)) and the α-TLHQ relative to α-CEHC ratio (−1% (95% CI: −13, 12)).

### 3.4. Sleep Quality and Vitamin E Serum and Urinary Levels

Poor sleep quality was associated with increased serum α-TOH levels (4% Beta (95% CI:1, 7)), but the effect size was small. No association was found between sleep quality and any urinary metabolite levels (α-TLHQ: 0% Beta (95% CI: −10, 10), α-CEHC: 0% Beta (95% CI: −12, 13), α-TLHQ relative to α-CEHC ratio: 1% Beta (95% CI: −9, 9)) (Table 2 and Appendix A).

### 3.5. Physical Activity and Vitamin E Serum and Urinary Levels

Compared with the lowest quartile of physical activity during leisure time, medium and high physical activity showed no different mean level of α-TOH (medium PA: −1% Beta (95% CI: −5, 3), high PA: −3% Beta (95% CI: −7, 9)). For urinary vitamin E metabolites, we observed a u-shaped association with physical activity intensity. Notably, the TLHQ relative to CEHC ratio was lower in the second quartile (−17% (95% CI: −26, −6]) and third quartile (−12% (95% CI: −22, −1)) compared to the reference (Table 2). However, there was no difference in TLHQ relative to CEHC ratio and any urinary vitamin E metabolite level between the highest quartile of physical activity and the lowest quartile.

### 3.6. Alcohol Consumption and Vitamin E Serum and Urinary Levels

We did not observe an association between alcohol intake and α-TOH level in serum. However, after adjusting for age, sex, BMI and smoking, alcohol consumption in the highest quartile was associated with a 17% lower α-TLHQ (−17% (95% CI: −28, −4)), as well as a lower TLHQ relative to CEHC ratio (−13% (95% CI: −24, −1)) compared to the lowest quartile group (Table 2 and Appendix A).

## 4. Discussion

For this study, we aimed to investigate the associations between lifestyle factors with serum α-TOH and urinary α-TOH metabolite levels in a cross-sectional analysis in a middle-aged population. After adjusting for possible confounding factors such as BMI, we observed associations between smoking, alcohol use and physical activity, and urinary α-TOH metabolite levels, whereas no associations were observed for serum α-TOH levels. Additionally, the Pearson correlation showed that serum α-TOH and urinary metabolites do not seem correlated to each other, suggesting both might reflect different biochemical processes. Especially given that observations were independent of BMI or total body fat, this indicated that the relation between certain lifestyle factors and urinary vitamin E metabolites are independent on an adverse adiposity profile. For sleep quality, we found an association with α-TOH, but not with urinary vitamin E metabolite levels. We did not find any association between diet quality and vitamin E levels in serum or urine.

Regarding the association between smoking behavior and vitamin E levels, a strong association was observed between current smoking and α-CEHC. This is in accordance with multiple other studies, all linking smoking to various measurements of increased oxidant activity [25,39,40,41]. Although our data showed that both the oxidative as well as enzymatic conversion of α-TOH were lower in current smokers, our data does suggest that there is relatively more oxidative conversion as enzymatic conversion, which is reflected by higher TLHQ relative to CEHC levels in current smokers. This suggests that there is a higher scavenging effect of oxidized lipids by α-TOH in current smokers. Additionally, we found a dose–response relationship in which current smokers with a higher number of pack years had high α-TLHQ relative to α-CECH and similar results were observed using the objective measure of current smoking cotinine. This supports the possibility of a direct effect of smoking on vitamin E metabolism. Furthermore, this may point to a higher amount of peroxidation, as well as lower antioxidant response capability in smokers than in non-smokers. The absence of this association in former-smokers suggests that smoking cessation reverses this observed difference. However, we acknowledge that our observations are cross-sectional, and longitudinal studies on smoking cessation are required to better understand the directional effects of smoking on the metabolism of vitamin E.

There was no association between diet quality and vitamin E levels. The DHD-index is a validated index and lower scores have been associated with an increased risk of chronic diseases [33]. The DHDI includes intake of the amount of total fat and saturated fat that theoretically can be associated with lipid levels or lipid peroxidation. A possible explanation for the lack of an association with vitamin E (metabolites) is that the DHDI is a reflection of multiple different food components, which might not all be relevant to lipid peroxidation. Consequently, two individuals may have the same DHDI score, but a very different habitual intake of the food components that are relevant for our study. In addition, we also acknowledge the potential measurement error in the questionnaire-derived data.

We did observe an association between poor sleep quality, indicated as a PSQI above 5, and higher α-TOH levels. As sleep deprivation is known to be associated with higher oxidative stress levels, we hypothesized an association between poor habitual sleep quality and α-TOH metabolite levels [42]. Although we found an association between poor sleep quality and higher α-TOH levels, poor sleep quality was not associated with urinary vitamin E metabolite levels. One study has shown that increased vitamin E levels can be associated with a reduction in sleep-deprivation induced problems [43]. However, additional studies are required to fully understand the relation between sleep disturbances and Vitamin E (metabolism).

Our finding that alcohol consumption is associated with a lower α-TLHQ is not in line with earlier studies. Lipid peroxidation can be a consequence of ethanol-induced oxidative stress, especially in the brain [44]. A reason for this discrepancy could be that there are other confounding factors that we were not able to control for that are linked to both alcohol consumption and a lower α-TLHQ. Additional studies would be warranted to study this possible relation in more detail.

The observed U-shaped association between physical activity and α-TOH metabolite levels is in accordance with previous literature, as physical activity has shown to be effective at increasing oxidant resistance [26]. When comparing the highest physical activity quartile with the lowest physical activity quartile, the ratio of α-TLHQ relative to α-CHEC level was not different. A potential explanation for this finding could be that during heavy physical activity, the body actually starts activating fat cells, which leads to an increase in lipid peroxidation. In turn, the high physical activity may cancel the beneficial effect of physical activity on the process [26].

One of the strengths of our study is that where most studies only investigated health outcomes in relation to serum α-TOH levels, or studied vitamin E supplementation as an intervention, we also assessed urinary α-TOH metabolites. These metabolites are thought to give a better representation of the enzymatic and non-enzymatic activity as they are the breakdown products of the ROS scavenging process. Additionally, we were able to do these measurements in an adequately sized population of 530 participants.

There are also several limitations of this study that need to be considered. Firstly, although this research was able to provide some novel insights on how α-TOH metabolism is associated with certain lifestyle factors, the cross-sectional design and observational nature of the data do not allow to determine whether the lifestyle factors preceded the changes in α-TOH metabolite levels. Secondly, α-TOH is only one of many antioxidants present in the human body. The full antioxidant defense system might also change with lifestyle, thus only assessing vitamin E metabolism might not be a complete representation of the functional antioxidant defense system. However, we did try to link the investigated lifestyle factors to lipid peroxidation, of which vitamin E is the main antioxidant response [2]. Lastly, all of the data on lifestyle factors were collected through questionnaires. With the exception of smoking, which we also studied using the metabolite cotinine, the other observations could have been influenced by recall bias and/or measurement error. Additionally, it is generally accepted that some lifestyle factors are frequently underreported, which specifically includes dietary intake and alcohol consumption. However, as we expect the underreporting is relatively independent from the vitamin E levels in both serum and urine, we do not expect this would majorly impact our observations.

## 5. Conclusions

In conclusion, our study suggests that, in our study population of relatively healthy middle-aged Dutch participants, the lifestyle factors smoking behavior, physical activity and alcohol consumption were associated with urinary vitamin E metabolites. Moreover, sleep quality was associated with serum α-TOH. These findings specifically highlight that measuring only vitamin E in serum does not provide sufficient insight in the antioxidant vitamin E capacity and activity. Alternatively, this might also explain why targeting antioxidant capacity only by increasing vitamin E levels in serum does not yield significant reduction in disease risk; targeting the conversion of vitamin E might provide additional health benefit beyond serum vitamin E concentrations. However, additional studies are required to provide further evidence of this hypothesis.

## Figures and Tables

**Figure 1 antioxidants-09-01280-f001:**
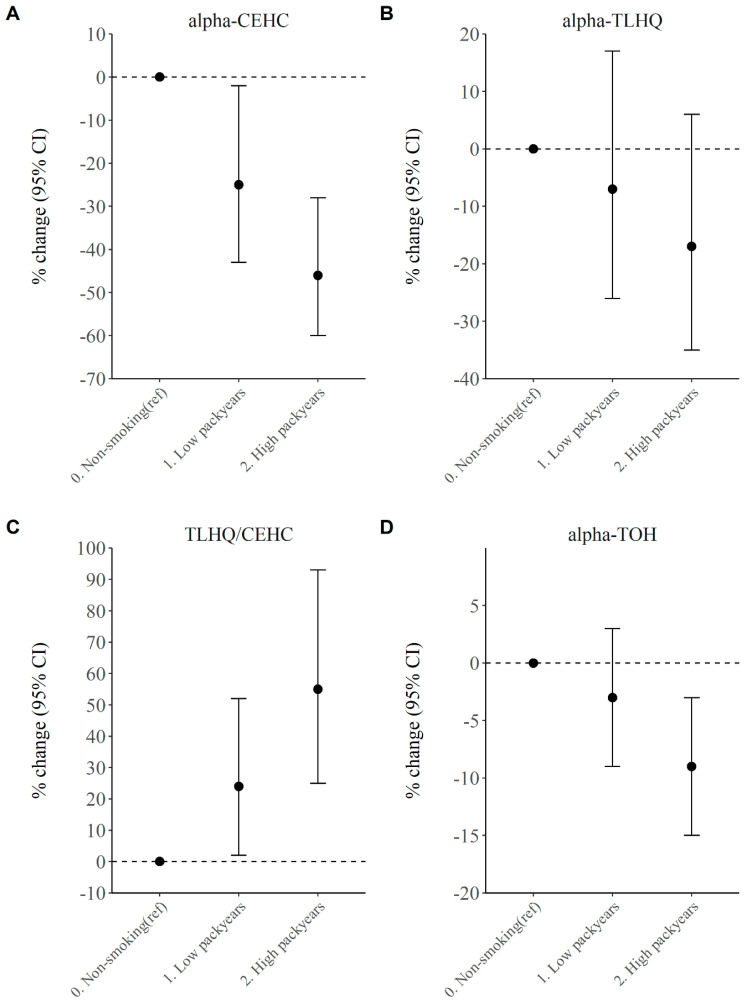
(**A**) Association between smoking intensity and urinary vitamin E metabolite α-CEHC in current smokers. (**B**) Association between smoking intensity and urinary vitamin E metabolite α-TLHQ in current smokers. (**C**) Association between smoking intensity and urinary vitamin E metabolite α-TLHQ relative to α-CEHC in current smokers. (**D**) Association between smoking intensity and urinary vitamin E metabolite α-TOH in current smokers. Results are derived from linear regression model adjusted for age, sex, BMI and alcohol with 95% confidence interval (CI) and presented as percentage difference compared with the reference group. Of the current smokers, the 50% with the lowest pack years is considered the low pack years group, whereas the highest 50% is in the high pack years group. CEHC, carboxymethyl-hydroxychroman.

**Table 1 antioxidants-09-01280-t001:** Characteristics of the Netherlands Epidemiology of Obesity participants stratified by sex.

	All (N = 530)	Men (N = 246)	Women (N = 284)
**Demography**	55.9 (6.0)	-	-
Sex, % men	46.4	-	-
Age (years), mean (sd)	55.9 (6.0)	56.2 (6.2)	55.6 (5.8)
**Lifestyle factors**	-	-	-
BMI (kg/m^2^), mean (sd)	25.9 (4.0)	26.6 (3.3)	25.4 (4.4)
Smoking, N (%)	-	-	-
Never	216 (40.8)	92 (37.4)	124 (43.7)
Former	255 (48.1)	124 (50.4)	131 (46.1)
Current	59 (11.1)	30 (12.2)	29 (10.2)
Pack years ^1^, median (IQR)	18.1 (8.4, 29.8)	26.6 (14.0, 36.4)	13.3 (4.7, 22.7)
Alcohol use (g/day), median (IQR)	9.2 (2.5, 21.5)	16.7 (5.1, 28.4)	7.2 (1.0, 14.4)
Leisure activity (MET-h), median (IQR)	30.0 (16.5, 49.6)	30.0 (16.0, 50.0)	29.9 (16.5, 47.8)
Poor Sleep Quality, N (%)	205 (38.7)	76 (30.9)	129 (45.4)
Diet Quality (0–130), mean (sd)	72.0 (14.6)	68.4 (13.4)	75.2 (14.9)
Lipid lowering drugs, N (%)	38 (7.2)	25 (10.2)	13 (4.6)
**Vitamin E metabolites**	-	-	-
**Measurements, median (IQR)**	-	-	-
α-THLQ (nmol/mmol creatinine)	1832 (1337, 2745)	1519 (1194, 2256)	2091.6 (1470, 3074)
α-CEHC (nmol/mmol creatinine)	265 (180, 439)	223 (137, 352)	307 (213, 508)
α-TLHQ/α-CEHC	2.0 (1.6, 2.3)	2.0 (1.7, 2.4)	2.0 (1.6, 2.3)
α-TOH	3.5 × 10^8^ (3.2 × 10^8^, 3.9 × 10^8^)	3.5 × 10^8^ (3.2 × 10^8^, 3.8 × 10^8^)	3.5 × 10^8^ (3.1 × 10^8^, 3.9 × 10^8^)

Data are presented as mean (standard deviation) or median (interquartile range, IQR)) for numerical variables, and number (proportions) for categorical variables. ^1^ Pack years only measured in current smokers. BMI, Body Mass Index.

**Table 2 antioxidants-09-01280-t002:** Associations between lifestyle factors and different measurements of vitamin E activity.

Lifestyle Factors		α -TOH		Total TLHQ		Total CEHC		Ratio TLHQ/CEHC	
	N =	% Change	95% CI	% Change	95% CI	% Change	95% CI	% Change	95% CI
Smoking	-	-	-	-	-	-	-	-	-
Nonsmokers	216	Ref	Ref	Ref	Ref	Ref	Ref	Ref	Ref
Former smokers	255	−1	−4 to 2	4	−6 to 15	2	−10 to 15	2	−7 to 12
Current smokers	59	−4	−8 to 0	−11	−24 to 5	−32	−44 to −18	32	14 to 53
DHDI	-	-	-	-	-	-	-	-	-
25.4–61.5	132	Ref	Ref	Ref	Ref	Ref	Ref	Ref	Ref
61.9–72.5	133	−2	−3 to 4	0	−13 to 15	6	−10 to 25	−7	−18 to 4
72.5–81.5	133	1	−2 to 5	13	−2 to 29	6	−10 to 25	5	−7 to 18
81.6–117.8	132	3	−1 to 7	12	−3 to 29	10	−7 to 31	−1	−13 to 12
Sleep Quality	-	-	-	-	-	-	-	-	-
Good Quality	288	Ref	Ref	Ref	Ref	Ref	Ref	Ref	Ref
Poor Quality	205	3	1 to 6	0	−10 to 10	-0	−12 to 13	1	−9 to 9
PA in MET-h/wk									
0.0–16.3	129	Ref	Ref	Ref	Ref	Ref	Ref	Ref	Ref
16.5–29.8	130	1	−2 to 5	−7	−18 to 7	11	−5 to 31	−17	−26 to −6
30.0–49.5	132	−1	−4 to 3	−11	−22 to 2	0	−14 to 18	−12	−22 to −1
49.8–242.5	130	−2	−6 to 1	−5	−17 to 14	4	−11 to 23	−9	−19 to 3
Alcohol use in g/day									
0–2.49	132	Ref	Ref	Ref	Ref	Ref	Ref	Ref	Ref
2.5–9.2	133	−2	−5 to 2	−8	−19 to 6	−1	−16 to 16	−8	−18 to 4
9.2–21.5	133	−0	−3 to 4	−9	−21 to 4	2	−14 to 20	−12	−22 to 0
21.5–171.6	132	3	−1 to 7	−17	−28 to 4	−4	−20 to 14	−13	−24 to −1

Results are derived from linear regression model adjusted for age, sex and BMI with 95% confidence interval (CI) and presented as percentage difference compared with the reference group. Analysis on smoking were additionally adjusted for alcohol use. Analysis on diet quality, sleep quality and alcohol use were additionally adjusted for smoking. Analysis on PA was additionally adjusted for smoking and alcohol. DHDI, Dutch Healthy Diet Index; PA, physical activity; TOH, tocopherol; TLHQ, tocopheronolactone hydroquinone; CEHC, carboxymethyl-hydroxychroman.

**Table 3 antioxidants-09-01280-t003:** Associations between cotinine and different measurements of vitamin E activity.

Cotinine Metabolite		α -TOH		Total TLHQ		Total CEHC		Ratio TLHQ/CEHC	
	N =	% Change	95% CI	% Change	95% CI	% Change	95% CI	% Change	95% CI
Cotinine	-	-	-	-	-	-	-	-	-
Absent	412	Ref	Ref	Ref	Ref	Ref	Ref	Ref	Ref
Present	118	2	−2 to 5	−11	−21 to −1	−20	−30 to −8	11	0 to 23
Cotinine levels	-	-	-	-	-	-	-	-	-
Absent	412	Ref	Ref	Ref	Ref	Ref	Ref	Ref	Ref
Low	59	5	0 to 10	−6	−19 to 10	−11	−26 to 7	6	−7 to 21
High	59	−2	−6 to 3	−16	−29 to −3	−28	−40 to −14	16	1 to 33

Results are derived from linear regression model adjusted for age, sex, BMI or total body fat and alcohol use with 95% confidence interval (CI) and presented as percentage difference compared with the reference group. Of the group with cotinine levels present, the 50% with the lowest levels is considered the low group, whereas the highest 50% is in the high group. α-TOH, α-tocopherol; TLHQ, tocopheronolactone hydroquinone; CEHC, carboxymethyl-hydroxychroman

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
