# Peer review of "Associations between Lifestyle Factors and Vitamin E Metabolites in the General Population"

_antioxidants, 2020, doi:10.3390/antiox9121280_

Round 1
Reviewer 1 Report
An article “Associations between lifestyle factors and vitamin E metabolites in the general population” written by L.G. Martens examines associations of serum α-TOH levels in fasting serum samples and vitamin E metabolites in urine with the main lifestyle factors smoking, sleep, physical activity, and habitual food and alcohol intake in a cross-sectional design. The paper and discussion is very interesting. However I have few remarks and questions.
- Could you explain why in Figure 2 only the result for CEHD but not for α-TOH, TLHQ, TLHQ relative to CEHC ratio is presented?
- The analysis of smoking intensity measured in pack years was restricted to never and current smokers. Could you explain why past smokers were excluded? Some authors include e.g. former-smokers who had quit within 5 years, as the influence of smoking in the past declines with time from quitting. Maybe including the group of past smokers who quit smoking at distant past is also possible. Please discuss pros and coins of different attitudes. Maybe such additional analyses could shed light on the effect of cessation of smoking, especially in the context of conclusion in discussion that “The absence of this association in former-smokers, suggests that smoking cessation reverses this observed difference”.
- In supplementary materials – under some tables should be stated that adjustment for sex was conducted only in analyses for “all” participants.
- Please extend the shortcut TBF (total body fat) used under the tables.
- In statistical analysis part was stated that “a multivariable linear regression analyses adjusted for age and sex” were performed for each individual lifestyle factor and vitamin E metabolite levels. “Additionally, we adjusted for BMI and performed a sensitivity analysis where we additionally adjusted for total body fat instead.” But under tables in manuscript is stated that “Results are derived from linear regression model adjusted for age, sex, BMI or TBF”. So which analyses were adjusted for BMI and which for TBF? Are presented somewhere the results of sensitivity analysis for alternative adjustment?
- Line 30: „age of 56(6)” please add the unit „years”.
- In Table 1maybe instead of “Age, years (SD)” should be used “Age [years], mean (sd)”
- Is α-TOH the acronym of α-tocopherol? Maybe consequently in tables instead of full nameα-tocopherol should be α-TOH used?
- Sometimes are used terms of “moderately associated” or weakly. Did you use defined cut-off points for moderate or weak association?
- Under Table 2 “Diet quality, sleep quality and alcohol use were additionally adjusted for smoking” is unclear. Maybe “Analyses on diet quality, sleep quality and alcohol use were additionally adjusted for smoking”
- In Figure 1 please replace “a” by “alpha”.
- In my opinion the sentence “As this observation could also be the result of Chance” is risky. Other associations also could be the result of chance.
Author Response
REPLY TO REVIEWER COMMENTS ON ANTIOXIDANTS-1017686
We thank the reviewer for the time and efforts to review our manuscript submitted to Antioxidants. We thank the editor for the willingness to reconsider a revised version of our manuscript for publication. Please find below point-by-point answers to the comments raised by the reviewer. Changes have been highlighted in yellow in the revised version of the manuscript.
Thank you for considering our revised manuscript for publication in Antioxidants.
Sincerely,
On behalf of the authors,
Leon G Martens MSc
Diana van Heemst PhD

Reviewer 2 Report
In this manuscript entitled “Associations between lifestyle factors and vitamin E metabolites in the general population,” the authors determined vitamin E and its metabolites in the serum and urine and investigated the relations to lifestyle factors. This manuscript includes interesting observations; however, some descriptions were inadequate and included over-interpretation. My major comments are as follows: Major comments: 1. In this manuscript, the authors evaluated the determinants by using means and 95%CI. The authors should discuss these by using other methods, i.e., statistical significance, correlation of each value, and ROC analysis, etc. The authors should show these precisely. 2. The value of alpha-tocopherol (α-T) in the serum is usually presented by α-T per serum cholesterol. The authors should evaluate by using α-T/Ch and compare with other values. 3. The statement “Examining serum α-TOH does not provide complete insight in vitamin E activity.” includes overestimation of the results. What is vitamin E activity? Can urinary α-T metabolites completely provide this activity? 4. The abbreviations without explanation in the abstract were not kind to readers. 5. In the introduction section, the authors did not summarize related previous reports precisely. The authors should show these and describe the significance of the purpose of the present study clearly. 6. In Table 1, the values of α-T lacked. 7. The results of Fig.1 are interesting. Please show it with the determinants of other vitamin E metabolites. 8. For example, the statement “Poor sleep quality was weakly associated with increased serum α-TOH levels (4% Beta [95% CI:1, 7]). No association was found between sleep quality and any urinary metabolite levels (α-TLHQ: 0% Beta [95% CI: -10, 10], α-CEHC: 0% Beta [95% CI: -12, 13], α-TLHQ relative to α-CEHC ratio: 1% Beta [95% CI: -9, 9]) (Table 2).”, I can not see this result and evaluate this statement precisely. The authors should show all results that readers can see.
Author Response

(The authors gave the same response as above.)

Round 2
Reviewer 2 Report
I have no concerns about this manuscript.